# Longitudinal Baboon (*Papio anubis*) Neutrophil to Lymphocyte Ratio (NLR), and Correlations with Monthly Sedation Rate and Within-Group Sedation Order

**DOI:** 10.3390/vetsci11090423

**Published:** 2024-09-11

**Authors:** Sarah J. Neal, Steven J. Schapiro, Elizabeth R. Magden

**Affiliations:** Department of Comparative Medicine, The University of Texas MD Anderson Cancer Center, Michale E. Keeling Center for Comparative Medicine and Research, 650 Cool Water Drive, Bastrop, TX 78602, USA; sschapir@mdanderson.org (S.J.S.); ermagden@mdanderson.org (E.R.M.)

**Keywords:** welfare, nonhuman primate, stress, immune dysregulation

## Abstract

**Simple Summary:**

There are many instances in which behavioral management refinements have impacted veterinary practices (e.g., positive reinforcement training and cooperative blood collection). There are fewer examples of veterinary information impacting behavioral management practices. We propose that the neutrophil to lymphocyte ratio (NLR), a cheap, simple, and easy marker of stress and inflammation often used in human clinical practice, has welfare utility in nonhuman primates (NHPs). However, there are only 7 published studies that have specifically examined NLR in NHPs, finding associations with longevity, certain medical conditions, and stressful circumstances. Therefore, we examined NLR as a function of stress resulting from clinical and research practices (i.e., sedation order within a group and sedation rate over time). We found that baboons sedated later within a group showed significantly higher NLR than those sedated earlier in the process, perhaps indicating an acute stress response. However, baboons with higher sedation rates per month showed lower NLRs, perhaps a reflection of more chronic stress. This points to the potential utility of NLR as a veterinary measure indicating possible dysregulation as a function of stress, and the opportunity for veterinarians and behavioral managers to work together to minimize stress through refinements to the environment, clinical care, and research practices.

**Abstract:**

Neutrophil to lymphocyte ratio (NLR) is a simple marker of stress and inflammation, but there is limited research regarding NLR in nonhuman primates (NHPs), with studies showing associations with longevity, certain medical conditions, and stressful circumstances. Here, we examined baboon NLR longitudinally, and as a function of health parameters. We also examined whether NLR was affected by sedation rate, as well as the order of sedation within a group, given that sedation events during clinical and research practices can induce stress in NHPs. While older adult and geriatric baboon NLR did not differ longitudinally, juvenile and young adult NLR tended to increase, primarily driven by increases in females. Additionally, baboons sedated later within a group showed significantly higher NLRs than those sedated earlier in the process. However, baboons with higher sedation rates per month showed lower NLRs. These data indicate that NLR may be dysregulated in different ways as a function of different types of stress, with sedation order (i.e., acute stress) causing pathological increases in NLR, and sedation rate over time (i.e., chronic stress) causing decreases. Importantly, we propose that NLR, a routinely obtained veterinary measure, has potential utility as a welfare indicator of stress resulting from clinical and research practices, as well as a measure that can inform behavioral management practices and interventions.

## 1. Introduction

Behavioral management often informs veterinary care practices in order to maximize the welfare of captive nonhuman primates (NHPs). In promoting and implementing practices like positive reinforcement training, socialization, enhanced enrichment, and creating functionally appropriate captive environments, behavioral management has, in many cases, reduced stress associated with routine veterinary clinical care and research practices (e.g., sample collection, physical examination, wound care, sedation, drug administration) [1]. On the other hand, veterinary information may also inform behavioral management practices. Information gained during veterinary care, such as routine hematological or serum chemistry, or even alopecia assessments [2], may indicate a need for behavioral management interventions or refinements. Furthermore, assessments of pre- and post-intervention welfare can use veterinary-based measures as dependent variables, illustrating just one example of the reciprocating pattern of the ways that veterinarians and behavioral scientists collaborate to manage and enhance the welfare of captive NHPs. However, physical parameters that are commonly used by veterinarians in routine monitoring of NHP health are not necessarily regularly used in behavioral management assessments regarding animal welfare. Here, we propose a measure that is routinely obtained and monitored by veterinarians with possible utility in welfare monitoring: neutrophil to lymphocyte ratio (NLR). Although NLR is commonly used in human clinical research and medicine with some association with stressful events, there are very few published studies on NLR in NHPs. Therefore, we aimed to examine NLR longitudinally as well as relationships between NLR, sedation events, and health parameters.

In human clinical medicine, NLR is an established tool used by clinicians in the screening, diagnosis, and prognosis of various diseases and pathological states. Normal NLR ranges between 1 and 3, with values above 4 or very low or persistently low NLRs (below 0.7) indicating poor prognosis, increased mortality, age-related disease, and other deleterious outcomes [3,4]. NLR has a relatively short history, with the first articles advocating for its use in human clinical practice published in 2001 [5]. Today, a PubMed search with “neutrophil to lymphocyte ratio” or “NLR” in the title or abstract yields more than 17,000 results, with the number of publications increasing annually since 2005, more than 1000 publications yearly since 2017, and almost 3000 publications in 2023 alone. Topics range from normal reference intervals to associations with mortality and associations with a variety of diseases and disorders (e.g., sepsis, appendicitis, tumors, cardiovascular disease, epilepsy, various forms of cancer, metabolic syndrome, stroke, brain injury, cirrhosis, pulmonary fibrosis, and multiple sclerosis, to name a few from just the top 30 publications from the PubMed search), to associations with psychological disorders and stress (e.g., history of childhood trauma or emotional abuse [6]).

While the search mentioned above yields results from across a variety of species (e.g., humans, horses, dogs, cats, rabbits, rats, and pigs), a PubMed search with the same criteria but the addition of “primate” or “monkey” yields just seven results. Given that nonhuman primates (NHPs) are often used as models for human diseases, disorders, and treatments due to their close phylogenetic relationship to humans, it is surprising that there are so few studies examining NLR across NHP species. More surprising is that there are numerous studies examining NLR in other nonhuman animals in the contexts of welfare, disease diagnosis and prognosis, and aging (e.g., [7,8,9,10]). Furthermore, given that hematology is standard practice during NHP physical exams in laboratory, zoo, and sanctuary settings, these data are often easily accessible. Therefore, it is unknown why NLR has not been explored more widely in NHPs.

To date, only seven studies exist regarding NLR in NHPs (Table 1). Consistent with the human literature, higher NLR in chimpanzees seems to be correlated with increased mortality, as chimpanzees with higher NLRs died at younger ages [11]. While the human literature shows increases in NLR with age (posited to be the result of age-related diseases), chimpanzee and baboon data seem to show a quadratic relationship with age, with higher NLRs found in middle-aged chimpanzees and young-adult baboons [11,12]. The authors hypothesize that this may be due to selective survival, such that individuals (particularly males) with higher NLRs die at younger ages, leaving a population of older individuals with lower NLRs [11,13]. In chimpanzees, males seem to have higher NLRs than females [11,14]. However, female baboons and rhesus macaques seem to have higher NLRs than males [15]). Across all species of NHPs studied thus far (i.e., chimpanzees, rhesus macaques, and baboons), mother-reared individuals tend to have higher NLRs than their nursery-reared counterparts [11,12,15]. Although several explanations have been posited for this relationship between rearing and NLR [12,15,16], the reasons have yet to be empirically investigated.

Of the seven published studies examining NLR in NHPs, five report data specifically regarding NLR’s association with stressful circumstances [12,15,17,18,19]. Both neutrophils and lymphocytes are known to change in response to stress, as they are both affected by circulating levels of glucocorticoids [20]. The association between NLR and stress may be mediated by cortisol, such that increased cortisol results in a reduced lymphocyte count and, in turn, increases the ratio between neutrophils and lymphocytes [18]. Under stressful circumstances, catecholamine and glucocorticoid secretion increase adrenaline in the system, which is followed by increased mobilization of neutrophils and destruction of lymphocytes (hence, increased NLR), while also increasing the half-life of neutrophils [17]. Indeed, a concurrent increase in cortisol and NLR has been found following chair training in female rhesus macaques [18], as well as following transport from China to Korea in cynomolgus macaques [17]. Even relocation to a different cage with a different social partner for 1–3 weeks caused an increase in NLR in macaques [19]. Some researchers are advocating for the use of NLR in human behavioral studies as indicators of mental health changes [20]. Given the association between NLR and stress in humans and other animals, including NHPs [21], NLR should have both clinical and welfare utility for NHPs. Indeed, we have previously shown that NLR is increased following transfer to a new facility, as baboon NLRs taken within one week post-transfer were significantly higher than NLRs taken during routine physical examinations one year later [12]. Additionally, we found significant elevations in post-transfer NLRs compared to pre-transfer levels in a subgroup of baboons. Given the results of these five studies, a positive association between transport stress and NLR seems to exist. However, only one study has investigated NLR as a function of other types of stress, such as research-related stressors, finding increased NLR following one month of chair-restraint training [18]. No studies to date have examined stress associated with routine clinical procedures, such as sedation for physical examinations.

The process of sedation can be a significant source of stress for NHPs, as it involves human access to and handling of the NHPs, injection, and a period of separation from the group [22,23]. Ketamine is often a preferred method of anesthesia for NHP sedation due to its safety, rapid effects, and relatively short duration of sedation. Although several studies have demonstrated changes in hematological parameters as a function of ketamine administration in NHPs, including changes in neutrophil and lymphocyte counts (but see Lugo-Roman, et al. [24]), the differences may be minimal compared to normal hematological variation among individuals [24,25,26,27,28,29]. Regardless, the effects of stress during chemical immobilization events cannot be teased apart from the effects of the ketamine, and, as a result, NLR obtained following ketamine administration may not be as accurate as NLR obtained without the use of ketamine (e.g., blood collected during leg presentation using positive reinforcement training [30]). However, given that all NLR data from our facility are derived from anesthetic events following ketamine administration, we can assume that the effects of ketamine on NLR are relatively constant, and therefore, that differences or changes in NLR following sedation events are at least partly reflective of the stress associated with such events.

The utility of NLR as an indicator of stress and health in NHPs has clinical, welfare, and behavioral management implications. A common goal of veterinarians and behavioral managers is to maximize the welfare of NHPs, and NLR may be a tool that can be used as part of an NHP’s welfare profile. For example, investigations into a high (or very low) NLR could lead to the identification of potential behavioral management interventions to alleviate stress or health issues in individual animals [12]. Similarly, procedures that seem to lead to or associate with changes in NLR across groups of animals could point to opportunities for larger-scale behavioral management interventions. However, before determinations can be made about the utility of NLR as a diagnostic, prognostic, and welfare tool in NHPs, additional verification of existing findings is needed to characterize NLR as a function of basic demographic factors, such as sex, rearing, and age. Furthermore, to date, the only longitudinal data that have been published in NHPs were in chimpanzees [11]. Given the strong relationships between NLR and health as well as NLR and stress in humans and animals, additional data are needed to examine such relationships in NHPs. The current study had four aims: (1) to present longitudinal data on baboon NLR; (2) to replicate previous findings regarding NLR and demographic variables of sex, rearing, and age using a larger sample; (3) to examine the relationship between health-related parameters recorded during physical examinations, including the presence of injury, pregnancy, and dependent infant status; and (4) to examine the association between NLR and stress using (a) sedation rate per month for either routine or research purposes, (b) assignment to study, and (c) order of sedation during routine physical exams within a large breeding group.

## 2. Materials and Methods

### 2.1. Subjects

Subjects included a total of 532 captive olive baboons (298 female, 234 male) housed at the Michale E. Keeling Center for Comparative Medicine and Research (KCCMR) at The University of Texas MD Anderson Cancer Center in Bastrop, Texas. The baboons ranged in age from 0 to 20.5 years (mean age = 4.10 years). The baboons were housed in corrals or Primadomes™ with inside access, or in indoor/outdoor runs, in three separate colonies on the KCCMR campus. The first colony is the specific pathogen free (SPF) colony (*n* = 319), the second is the conventional colony (non-pathogen-free; *n* = 47), and the third colony is the SPF nursery (*n* = 166). There were 256 mother-reared and 276 nursery-reared baboons. Nursery-reared individuals were defined as those baboons who were separated from the dam within 24 h following birth and cared for by humans and raised in an incubator with access to human infant formula until they were put into small, same-age peer social groups from 3 months until 2 years of age, when they were introduced to larger adult and sub-adult social groups. Mother-reared individuals were defined as baboons who were not separated from their dam for at least the first 6 months of life and were reared in their natal group during that time. Table 2 depicts the number of baboons across sex, rearing status, and age categories for the two studies/datasets (see below). Baboons were housed in breeding groups consisting of one or two breeding males, 12–16 breeding females, and in the SPF colony, their juvenile and infant offspring (0–3 years of age).

### 2.2. Data Collection

NLR values were taken from hematology records obtained during routine biannual physical exams. Blood samples were collected in BD Vacutainer^®^ EDTA tubes while animals were sedated with ketamine. To calculate NLR for each baboon, we divided percent values for neutrophils by percent values for lymphocytes, as we have done previously [11,12]. The age of the baboon at the time at which NLR values were obtained was used as the age variable. All research and experimental protocols complied with those approved by the UTMDACC Institutional Animal Care and Use Committee, as well as the legal requirements of the United States and the ethical guidelines put forth by the American Association for Laboratory Animal Science (AALAS), the Animal Welfare Act, and The Guide for the Care and Use of Laboratory Animals. The Keeling Center has been fully accredited (continuously) since 1979 by AAALAC.

Two datasets were used in this study. The first dataset consisted of NLR values taken between May 2017 (arrival of the baboons to our facility) and February 2023 (532 subjects included in the dataset). A report was generated from the KCCMR electronic medical record (EMR) database showing all dates that each baboon was sedated while it was housed on the KCCMR campus. For each baboon, we calculated an average number of sedations per month, which was equal to the number of sedations divided by the number of months elapsed between the first and last sedation. For animals born in 2017 or earlier, the first sedation was a quarantine exam upon arrival at the KCCMR facility. For animals born in 2018 or later, the first sedation was a physical exam between 1 and 6 months of age. For these animals, we entered the birth date as the “start date” to calculate months elapsed. We excluded any exams in which an injury or other medical condition was noted. Each baboon had between 1 and 8 NLR values in the dataset.

In dataset 1, we also noted whether baboons were included in a study (N = 66, primarily vaccine studies). We excluded NLRs directly following any experimental treatments but included a post-experiment NLR (i.e., NLRs taken following the conclusion of the study). These baboons had significantly higher sedation rates (mean = 1.05 per month, SEM = 0.06) than non-study baboons (mean = 0.30 per month, SEM = 0.01). Although non-study baboons were scheduled to be sedated only twice per year for biannual physical exams, other reasons for sedation included biologics requests (e.g., screening for potential sales, blood units, MRI scans), treatment for injuries, infant access, etc., thereby increasing the sedation rate for non-study baboons above two per year (which would be expected based on biannual exams) to approximately 3.7 per year. Study baboons were also significantly younger than non-study baboons. Therefore, we created a subgroup of non-study baboons who were matched to study baboons as closely as possible on age, rearing, and sex for further analyses.

The second dataset was used to examine NLR as a function of variables collected during routine physical exams between August and November 2023. During these routine physical exams, we recorded which animals were sedated first through last within each group for 231 baboons (165 female, 88 nursery-reared, mean age = 5.66 years) across 10 social groups. The process for sedation consists of care staff and trainers shifting baboons within their home enclosure through a tunnel system to briefly isolate the baboon in order to inject it with ketamine. As a result, animals who were sedated later in the process waited a longer period of time (up to 2 h) while being present during the other baboons’ sedations in the indoor portion of the home enclosure. We divided neutrophil percent by lymphocyte percent derived from clinical hematology taken during that sedation to create NLR.

This second dataset also included other variables of interest recorded during physical exams, including the presence of injury, pregnancy, and dependent infant status. During physical exams, the presence of an injury was recorded by the veterinarian. In the current study, these ranged from no injury noted on examination (*n* = 163), to minor injuries (small lacerations, scabs, previous healing wounding, *n* = 45), to moderate injuries (e.g., punctures, minor lacerations, abrasions, *n* = 19), to severe injuries (bites, lacerations, conditions that required analgesic treatment, *n* = 4). We also recorded whether females were pregnant (N = 112 adult females with complete data: not pregnant *n* = 73, pregnant *n* = 39), and whether they had a dependent infant at the time of the physical exam (no infant *n* = 68, dependent infant = 44). A dependent infant was defined as 6 months of age or younger, corresponding to weaning age.

### 2.3. Data Analysis

We used SPSS version 26 statistical software for all analyses (IBM, 2021). Data are available from the corresponding author. Histograms and Q-Q plots showed that the data were positively skewed. Therefore, we used a log10 transformation for all NLR outcome variables in our analyses (hereafter referred to as “lg10NLR”). Visual inspection of residuals and QQ plots of residuals by fitted values for log-transformed data showed good homoscedasticity. Unless otherwise noted, all analyses used lg10NLR values. We occasionally report raw NLR means, referring to non-transformed NLR values, to increase the interpretability of results. We used *p*-values of <0.05 to determine statistical significance, with *p* < 0.10 as trending.

#### 2.3.1. Longitudinal NLR

To examine longitudinal NLR in baboons (Table 3 analysis 1a), we used a repeated-measures analysis of covariance (ANCOVA) with sex and rearing as between-subject factors, age as a covariate, and lg10NLR for years 1 through 5 as the repeated measure. For this analysis, we selected baboons who were not part of a study and those who had 5 consecutive NLR values (N = 174). To further explore the effects of age, we used a curve estimation procedure (Table 3 analysis 1b) with age at the most recent NLR as the independent variable and the most recent lg10NLR as the dependent variable (again, excluding study animals). Note that, given the small number of males in our dataset, sex effects should be interpreted with caution.

#### 2.3.2. Sedation Rate Per Month

To examine the effects of sedation rate on NLR (Table 3 analysis 2a) we used a multiple regression with age at most recent NLR, sex, and rearing, entered on the first block of the equation, assignment to study entered on the next block, sedation ratio entered on the last block, and the most recent lg10NLR value as the dependent variable across the entire sample (N = 532). We then repeated this analysis using control (i.e., non-study) baboons matched with study baboons on age, sex, and rearing (Table 3 analysis 2b). This allowed us to examine whether assignment to study and sedation rate uniquely explained the variance in the most recent NLR. To examine whether the sedation rate changed NLR values over time, we calculated a change score by subtracting each baboon’s last NLR value from their first NLR value (Table 3 analysis 2c). We selected non-study baboons, as well as baboons who had a minimum of 24 months between NLR data points with which to calculate change scores. We used linear regression with sex, rearing, and age (age at most recently measured NLR) entered on the first block of the equation and sedation rate on the second block.

#### 2.3.3. Sedation Order

We used linear regression with sex, rearing, and age on the first block of the equation and sedation order on the second block to examine whether sedation order explained NLR over and above those demographic variables (Table 3 analysis 3).

#### 2.3.4. Health-Related Parameters

We examined relationships between NLR and the presence of injury recorded during routine physical exams using a univariate ANCOVA with lg10NLR as the dependent variable and the presence of an injury (categorized as present/absent due to unequal sample sizes in the severity classifications described above) as the independent variable, with sex and rearing as between-subject factors, and age as a covariate (Table 3 analysis 4). We then examined relationships between NLR and pregnancy and the presence of a dependent infant recorded during physical exams among adult females. We used a univariate ANCOVA with pregnancy (pregnant/not pregnant) and infant presence (no dependent infant/dependent infant 0–6 months of age) as independent variables, with age and rearing as covariates (Table 3 analysis 5).

## 3. Results

### 3.1. Longitudinal NLR

Sphericity was violated in the ANCOVA, so we report Greenhouse–Geisser corrected statistics, including exact degrees of freedom in decimals. There was a significant effect of time, F(3.76,635.68) = 9.48, *p* < 0.001, although pairwise comparisons revealed that no pairs of NLR were significantly different from one another (*p* > 0.11). There was also a significant effect of age [F(1,169) = 46.19, *p* < 0.001], rearing [F(1,169) = 3.99, *p* < 0.05], and a trending effect of sex [F(1,169) = 2.73, *p* = 0.10]. Females trended toward higher NLRs than males across years, and mother-reared individuals had higher NLRs than nursery-reared individuals (Figure 1).

The curve estimation further examining the effects of age on NLR showed a significant positive linear effect of age [F(1,471) = 6.03, *p* < 0.02, R2adj = 0.011], as well as a significant quadratic effect [F(2,470) = 10.91, *p* < 0.001, R2adj = 0.04]. Consistent with our previous publication [12], these data suggest that NLR is highest around 5–10 years of age (Figure 2).

### 3.2. Sedation Rate Per Month

Descriptive statistics showed that, among non-study baboons, sedation rates ranged from 0.08 to 1 per month (N = 466, mean = 0.30, SEM = 0.005, median = 0.29 per month). Among baboons assigned to the study, the sedation rate ranged from 0.43 to 2 per month (N = 66, mean = 1.06, SEM = 0.06, median = 1.07 per month).

The final model examining assignment to study and sedation rate across the whole sample was significant: F(5,526) = 39.38, *p* < 0.001, R2adj = 0.27. While assignment to study was not a significant predictor of NLR (*p* > 0.50), sedation rate was significant, beta = −0.48, t = −5.74, *p* < 0.001. Similarly, when using only control (i.e., non-study) baboons matched with study baboons on age, sex, and rearing, the final model showed that sedation rate significantly predicted NLR whereas assignment to study did not, F(5,125) = 16.46, *p* < 0.001 (see Table 4 for model coefficients). Although assignment to study was not a significant predictor of NLR (*p* = 0.50), sedation rate was (beta = −4.3, t = −4.5, *p* = 0.001), with the final model showing that age, sex, rearing, assignment to study, and sedation rate together explained 37% of the variance in NLR values. Unexpectedly, the sedation rate showed a negative relationship with NLR, such that baboons who were sedated more often showed lower NLRs (Figure 3). Raw means showed that study baboons’ NLR (mean = 1.46 ± 0.33) was half that of matched non-study baboons (mean = 3.28 ± 0.33).

When examining change scores in non-study baboons, we found that the final model was significant [F(4,232) = 6.29, *p* < 0.001], but the sedation rate did not predict NLR change scores (*p* > 0.36). Age was a significant predictor of change in NLR (beta = −0.22, t = −3.83, *p* < 0.001), as was sex (beta = −1.77, t = −2.58, *p* < 0.01), with females showing a higher NLR increase over time compared to males across all age categories (Figure 4b). To graphically depict the age and sex effects, we divided baboons into age categories based on their most recent age, and categorized change scores as increased (positive change scores ≥ 1 unit in the ratio), decreased (negative change scores ≤ −1), or no change (0 ± 0.99). As shown in Figure 4a, the majority of baboons who were considered juveniles (0–4 years of age, *n* = 91) and young adults (5–9 years of age, *n* = 57) at the conclusion of the sampling period showed increases in NLR over the sampling period, whereas the majority of baboons who were considered older (10–14 years of age, *n* = 50) or geriatric (15 years or older, *n* = 39) exhibited no change or decreases in NLR over the sampling period. Figure 4b plots raw change scores in box-and-whisker plots across age and sex categories, showing that some individuals exhibited large changes in NLR over the sample period, as indicated by the outliers in each age category. Figure 4c shows that, on average, juvenile and young adult baboons tended to show increases in NLR over time whereas older adults tended to show no change and geriatric baboons tended to show decreases in NLR over time, with females showing a higher NLR increase over time compared to males across all age categories (Figure 4d). It should be noted that the decrease seen in geriatric baboons over time is driven by the only two male geriatric baboons in the sample, which have change scores of −6 and −7.

### 3.3. Sedation Order

The linear regression was significant, F(4,226) = 19.52, *p* < 0.001, R2adj = 0.244. NLR values were higher in baboons who were sedated later in the group (beta = 0.19, t = 7.30, *p* < 0.001). Sex and rearing were both trending predictors of NLR in the model, with visual inspection of the graphs showing that the order effect was more pronounced in females (beta = −0.09, t = −1.75, *p* =0.08, Figure 5). Additionally, consistent with previous data, NLR was generally higher in mother-reared baboons (beta = −0.08, t = −1.77, *p* = 0.08). Age was not a significant predictor in this model (*p* = 0.15).

### 3.4. Health-Related Parameters

Only rearing and sex were significant in the analysis (*p* < 0.01). The presence of an injury was not a significant factor in the model (*p* > 0.41). However, there was a trending interaction between sex and injury (Figure 6A): NLR did not differ between non-injured males (raw NLR mean = 5.12 ± SD 3.61, *n* = 48) and females (raw NLR mean = 6.31 ± SD 4.91, *n* = 115), but injured females had higher NLRs (raw NLR mean = 7.31 ± SD 5.58, *n* = 50) than injured males (raw NLR mean = 4.63 ± SD 4.42, *n* = 18).

The main effects of pregnancy (*p* > 0.9) and dependent infant (*p* > 0.9) were not significant. However, there was a trending interaction between rearing and the presence of a dependent infant, F(1,103) = 3.24, *p* = 0.075 (Figure 6B). While mother- and nursery-reared females with no dependent infant had similar NLRs (raw NLR means: MR = 7.78 ± SD 6.51, *n* = 32; NR = 5.60 ± SD 5.00, *n* = 36), mother-reared females with dependent infants (raw NLR mean = 9.71 ± SD 4.83, *n* = 24) had higher NLRs than nursery-reared females with dependent infants (raw NLR mean = 4.61 ± SD 4.18, *n* = 20).

## 4. Discussion

This paper accomplished several aims. First, we replicated previous data showing that NLR is higher in mother-reared baboons and in females [12]. Second, we report the first longitudinal old-world monkey data, and only the second set of longitudinal data in NHPs overall. Third, we report the first data showing associations between NLR and stressful events not associated with transport. Lastly, we add evidence that NLR can be related to certain health-related parameters, such as injury and the presence of a dependent infant, and importantly, that these associations can be different across sex and rearing, respectively. It is important to note, however, that there is a small number of males in our sample given that these data are derived from a breeding colony in which groups are housed with just one or two males per group. As is the case in our previous study [12], sex, rearing, and age are confounded, such that the majority of the oldest baboons are mother-reared females. Therefore, although the data from the current study are consistent with our previous study, the main effects regarding sex and rearing should be replicated with a larger sample of older baboons, more males, and both mother- and -nursery-reared baboons across all age categories.

NLR has the potential to be a useful physiological indicator of well-being in NHPs given its association with stress, but a “normal range” or reference interval is needed before interpretations regarding “deviant” NLR can be made by veterinarians. Unfortunately, due to our small sample of males and geriatric individuals, as well as the differences between MR and NR baboons, we are hesitant to report “normal” ranges of NLR. However, we can make some descriptive statements about the dataset. We previously reported that 95% of NLR values in a sample of 387 baboons fell between 0.43 and 6.5, with a mean of 2.37 and median of 1.63 [12]. In the current study, across 532 subjects, the average NLR (i.e., the mean of NLR averaged over time within each baboon) was 3.20 (median 2.56), and 95% of the sample fell between 0.42 and 10.29. As a result, with a larger sample size using NLR averaged over time, we found a larger range and a higher mean and median, but a very similar lower value of NLR. It would be beneficial to calculate reference ranges for males and females, as well as across rearing statuses and age groups. In this way, veterinarians could potentially make statements regarding pathological vs. normal NLR in baboons. Once identified, the potential causes of such pathology could be investigated, and in the case of suspected environmental stressors, behavioral management teams could be alerted, and interventions developed to address the issue. This highlights the importance of collaborations between veterinary and behavioral management teams and demonstrates the impact of veterinary information on behavioral management.

In the current study, we aimed to provide the first longitudinal NLR data in an old-world monkey. We previously reported that individual chimpanzees show no change in NLR over 5- and 10-year timespans, but that the cross-sectional data showed differences as a function of age group, with middle-aged individuals showing higher NLR than young and old chimpanzees [11]. Baboons in the current study show a very similar pattern: the longitudinal data revealed no change over time within individuals, but the cross-sectional data showed differences in NLR as a function of age group, with young adults showing the highest NLRs. Furthermore, these data seemed to show that juvenile and young adult NLR increase over a 5-year timespan, mostly driven by increases in females in these age groups, whereas older adult and geriatric baboons tend to show no change over time. Interestingly, this is a remarkably similar pattern of results regarding accelerated age in baboons recently found by our group, wherein juveniles and young adults show accelerated age (i.e., DNA methylation age older than chronological age) but adult and geriatric baboons show no change and decelerated age, respectively [31]. Therefore, it seems that younger baboons show increases in NLR over time as well as accelerated age, while older baboons show no change in NLR and decelerated age [31].

Given that NLR has been shown to increase following stressful events (i.e., in response to various types of transfers and restraint training), we aimed to examine whether NLR changed as a function of sedation events. We defined two types of sedation events: (1) the sedation rate per month, and (2) the order of sedation within a group at one point in time. The sedation rate was calculated as the number of sedations per month, with our baboon sample having a mean of 0.30 sedations per month (i.e., equivalent to one sedation approximately every three months), and ranging from 0.08 to 1 per month among non-study baboons. This is a relatively low rate in comparison to studies examining the effects of ketamine sedation in NHPs, which often utilize a paradigm that involves daily sedation for at least one month [24,32,33,34,35]. Some studies use “chronic” or “acute” models of ketamine administration, which show widespread effects of sedation on behavior, brain structure and function, and cognition [32,33,34,35,36,37]. Therefore, it is possible that NLR was not affected by the sedation rate among non-study baboons due to a floor effect: the sedation rates in the current study were too low to detect an effect across the colony.

The idea of a floor effect is further supported by the finding that study baboons, which had a significantly higher sedation rate than non-study baboons, did show a change in NLR as a function of sedation rate. However, this result was in the opposite direction as hypothesized, given previous findings showing increased NLR as a function of other stressful events, such as transport [12,17,19]. Here, we found that study baboons showed lower NLR values after repeated sedations compared to age-, rearing- and sex-matched controls. Lower NLR may be similar to the dysregulation seen in human and NHP cortisol values, wherein both high and low (blunted stress response or altered set point) values indicate dysregulation [22]. Indeed, human studies point out that NLR values lower than 0.7 tend to indicate some type of pathological state or process [4]. Additionally, we found that assignment to study did not significantly contribute to NLR values, whereas sedation rate did. Together, these results suggest that the rate of sedations per month, rather than being assigned to a study per se, creates some type of dysfunction in neutrophils and lymphocytes, such that NLR is low following a higher rate of sedations per month over a period of several months. These lower NLRs associated with sedation rate may be an indication of chronic stress on the system, as this is consistent with the negative relationship between the risk of airway hyperresponsiveness, cortisol stress scores, and emotionality found in rhesus macaques [38]. These results point to the importance of including low NLR values (as well as very high values) as a risk factor for pathological states or processes in old-world monkeys. Importantly, this pattern of potentially pathological NLR that is seemingly related to veterinary and research practices points to the opportunity for veterinary and behavioral management teams to work together to refine such practices to decrease stress during research and clinical care procedures (see below; [1,30,39,40,41]).

We also examined sedation order in relation to NLR, defined as the order in which baboons were funneled through the tunnels and sedated on their group’s routine physical exam day. We found a positive correlation between NLR and sedation order, with higher NLR in baboons who were sedated later in the process, perhaps reflecting a short-term, acute stress response. The positive association may be akin to that found during transport stress. Although transport stress occurs on a longer-term basis (from a period of one day to up to one week), it also elicits a more acute rather than chronic stress response, with a 2-fold increase from baseline to post-transport across apes [42] and three species of old-world monkeys [12,17,19]. Similarly, we found that baboons who were sedated in the last 25% of the group exhibited NLR values that were twice as high (raw NLR mean = 9.64, SEM = 0.75) as those sedated in the first 25% of the group (raw NLR mean = 4.21, SEM = 0.57). From a welfare perspective, this perhaps points to the need to alter the current sedation process at our facility, and our veterinary and behavioral management teams are in the planning stages of modifying specific practices to potentially decrease stress during these group-wide sedation events. These modifications include providing visual barriers so those animals sedated later cannot observe the earlier animal sedations, as well as developing processes that would enable leaving portions of the group outdoors in their typical housing while the animals are being sedated, thus minimizing visual observation of sedations and also limiting the auditory exposure to sedation noises. Additionally, we plan to use positive reinforcement training to increase voluntary participation and cooperation during sedation procedures, which has been shown to result in decreased behavioral and physiological stress and increased well-being across NHP species [30,40,43]. We will then compare pre- and post-modification NLR to empirically evaluate the efficacy of such stress-reducing measures.

Lastly, we examined NLR as a function of health-related parameters that were also available to us from data recorded during routine physical exams. Unexpectedly, NLR did not differ as a function of injury when comparing baboons with and without an injury present. However, interestingly, we found an interaction for NLR between sex and injury, as well as between rearing and the presence of a dependent infant. When no injury was present, male and female baboon NLR were not significantly different. Similarly, without a dependent infant, mother- and nursery-reared female NLRs were not significantly different. However, when an injury was present, female NLR was higher than male NLR, and when a dependent infant was present, mother-reared NLR was higher than nursery-reared NLR. This perhaps suggests that in these male-dominated social hierarchies, female baboons have an increased level of stress associated with injury, as they are more susceptible to aggression from both conspecific males and females. The subsequent injury risk to the males is less, as they hold a more dominant social position. The results further suggest that the added stress and/or physiological demands of injury and the presence of a dependent infant exacerbate or highlight the difference in NLR between males and females and between MR and NR baboons. Under relatively normal conditions, the differences in NLR as a function of sex and rearing are perhaps less apparent (as can be seen descriptively in Figure 6, female and MR baboons still have slightly higher NLRs than their male and NR counterparts). However, when a system stressor is present, such as injury or infants, the differences in NLR become more apparent. Notably, we found that NLR did not differ as a function of pregnancy, another type of system stressor, which is inconsistent with our previous within-subject analysis that showed higher NLRs when females were pregnant compared to when they were not pregnant and that did not differ across trimester [12]. The current cross-sectional data showed no difference between pregnant and non-pregnant females, perhaps highlighting the additional power of within-subject analyses. Continued examination of NLR as a function of other types of system stressors will be important in further elucidating the relationship between NLR and physiological stress.

It is possible that changes in NLR are dependent on the type of stressor present. Results from the current study and Neal et al. (2024) potentially suggest that, in baboons, chronic stress (i.e., nursery-rearing, more sedations) results in lower NLR, whereas acute stress (i.e., transport and sedation order) results in higher NLR. We replicated the finding that nursery-reared baboons show lower NLRs than mother-reared baboons and previously discussed several possibilities for this finding [12], including the possibility of a blunted immune-inflammatory response that is similar to blunted cortisol responses following chronic exposure to stress [22]. Previous studies in rhesus macaques have found higher lymphocyte proliferation in response to mitogen stimulation in NR compared to MR monkeys [44], that NR monkeys have lower neutrophil counts, and indoor-reared monkeys, including NR monkeys, had higher lymphocyte counts (although this cannot be attributed specifically to rearing given that the indoor MR subjects also had elevated lymphocyte counts) [45]. Furthermore, NR rhesus macaques seem to be less physiologically responsive than MR monkeys, showing lower plasma cortisol concentrations following various stressors (e.g., housing changes, social separation, pharmacologic challenge [45,46,47,48]. However, hair cortisol concentrations, a measure of cortisol over a longer period of time, have been shown to be higher in NR compared to MR macaques, and NR macaques show higher increases in hair cortisol in response to relocation [49]. Other studies have suggested that nursery-rearing alters cortisol baseline levels, or “set points”, but may not affect acute responses to stressors [50]. These studies, although in rhesus macaques and not baboons, suggest that nursery rearing alters HPA and immune function and regulation [47]. As such, the data from the current study may lend evidence to the idea that lower NLR in nursery-reared baboons and as a function of more sedations over time (both chronic stressors) reflect the lower NLR “set point” and/or blunted NLR response. This hypothesis could be examined by obtaining cortisol and NLR at the same time to determine correlations among hair and plasma cortisol, NLR, and rearing.

## 5. Conclusions

Our longitudinal and cross-sectional data seem to mirror patterns shown previously in chimpanzees, rhesus macaques, and baboons; NLR does not seem to change over time within individuals but does show a quadratic relationship with age. This pattern may point to an effect of selective survival, such that individuals who have higher NLR (reflective of underlying pathophysiology) die at younger ages, leaving baboons with lower NLR living to older ages [11,12]. We replicated previous results showing that mother-reared baboons have higher NLR than nursery-reared baboons, and females show higher NLR than males, and, interestingly, these sex and rearing differences seem to be more apparent when certain types of additional stressors are present, including an injury or dependent infant. Lastly, NLR seems to reflect states of stress, as measured during sedation events. While chronic stress as a result of higher sedation rates may be reflected by a low NLR, short-term or acute stress as a function of sedation order within a single sedation event seems to result in increased NLR.

These results have behavioral management implications and highlight the opportunity for interfacing between veterinarians and behavioral management. For example, the fact that sedation order has a robust effect on NLR, likely indicating elevated stress, points to opportunities for behavioral management interventions, such as positive reinforcement training to increase voluntary participation in such clinical activities. The fact that NLR is consistently lower in baboons who are nursery-reared (an abnormal rearing condition) potentially suggests opportunities for behavioral management interventions, such as increased peer contact while in the incubator or different types of surrogates to increase the functional appropriateness of the nursery environment. The amount of individual variation in baboon NLR points to an opportunity for individualized care; a veterinarian who has identified a baboon with abnormally high NLR and suspects that the cause is stress-related could work with the behavioral management team to find the cause and appropriate interventions for that particular animal. Additionally, empirical evaluations of behavioral management interventions could include the use of NLR as a dependent variable and measure of stress to examine the success of these strategies.

## Figures and Tables

**Figure 1 vetsci-11-00423-f001:**
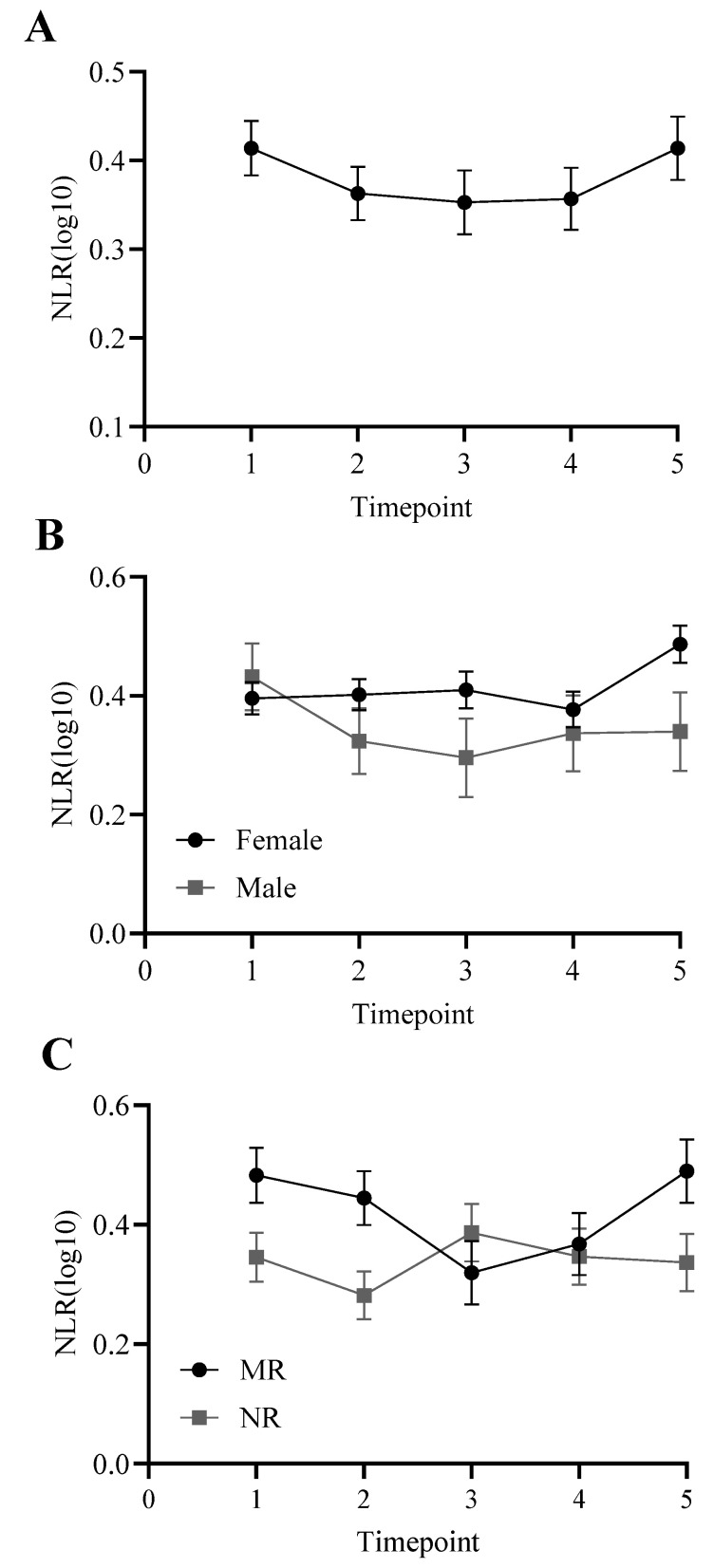
NLR across timepoints 1 through 5. Panel (**A**): collapsed across sex and rearing, with a significant effect of time but pairwise comparisons were not significant (*p* > 0.11); (**B**): trending effect of sex (*p* = 0.10); (**C**): significant main effect of rearing (*p* < 0.05): mother-reared (MR), nursery-reared (NR). Log10 refers to the log-transformed NLR variable.

**Figure 2 vetsci-11-00423-f002:**
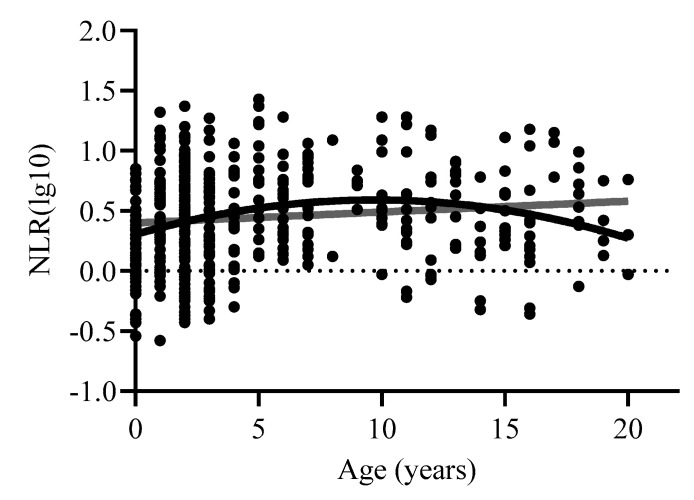
Relationship between NLR and age across the entire sample. Log10 refers to the log-transformed NLR variable.

**Figure 3 vetsci-11-00423-f003:**
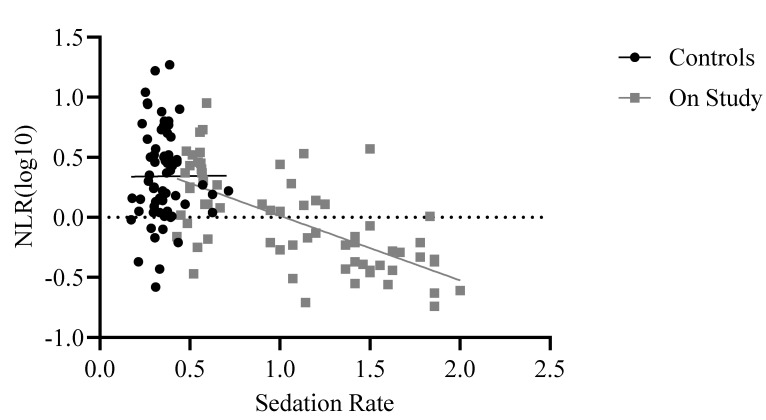
Relationship between sedation rate per month and NLR across baboons who were on study and control baboons who were matched on age, sex, and rearing. Control baboons (black trendline) showed no relationship between NLR and sedation rate, while study baboons (gray trendline) showed a significant negative relationship. Log10 refers to the log-transformed NLR variable. The dotted line indicates zero.

**Figure 4 vetsci-11-00423-f004:**
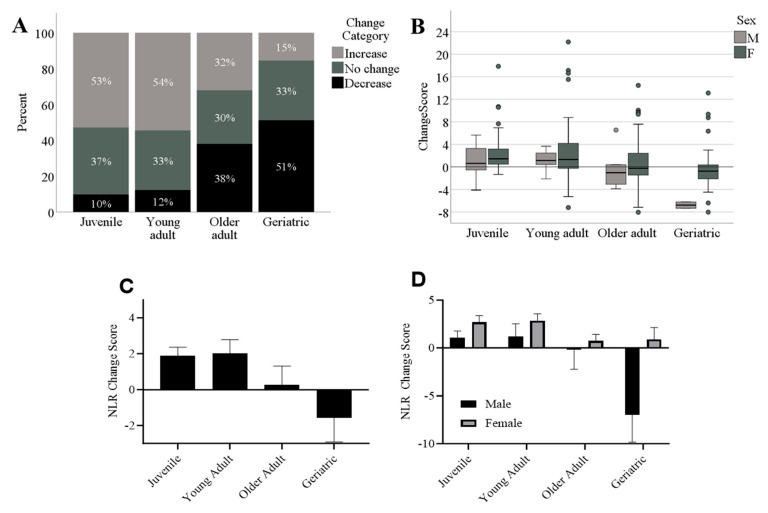
(**A**) The percentage of each age category showing increases (≥1 ratio unit increase), decreases (≥1 ratio unit decrease), or no change (0 ± 0.99 ratio unit change) in NLR over time. (**B**) Box-and-whisker plots showing change scores across age categories and sex. (**C**) Average change scores (NLR at first timepoint minus NLR at most recent timepoint) across age groups, showing that NLR increased over time in juvenile and young adults, but decreased over time for geriatric baboons. (**D**) Females tend to drive the increases found in juveniles and young adults over time, and the two only males tend to drive the decreases in geriatric baboons. M: males; F: females. NLR change scores based on raw NLR data for each baboon.

**Figure 5 vetsci-11-00423-f005:**
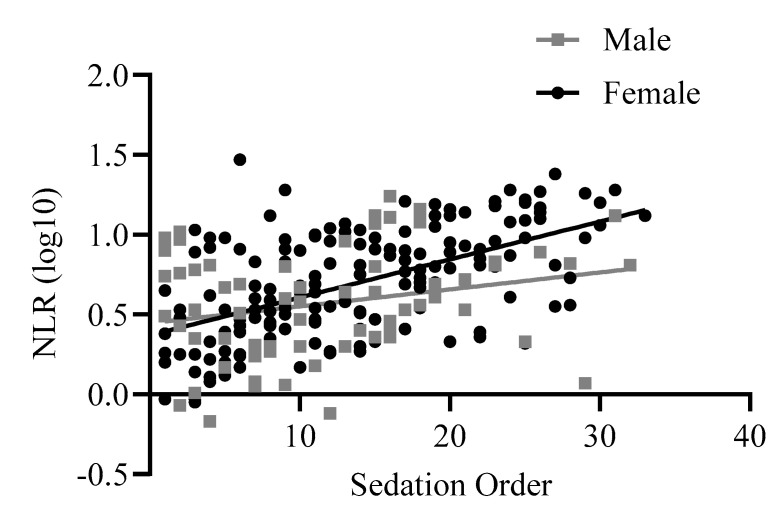
The linear relationship between NLR (log 10 transformed) and sedation order across males and females. Log10 refers to the log-transformed NLR variable.

**Figure 6 vetsci-11-00423-f006:**
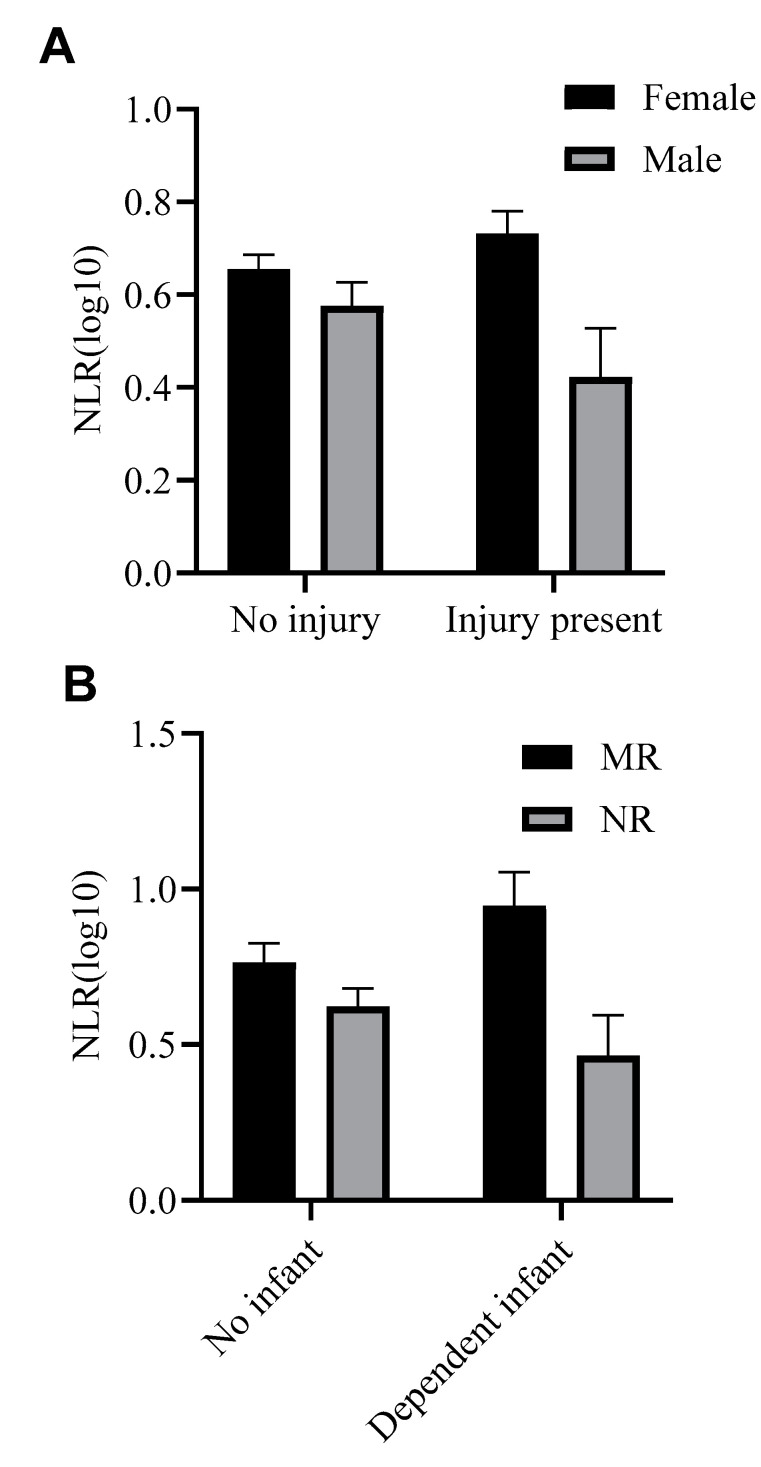
(**A**): Differences in mean NLR (log 10 transformed) as a function of sex and presence of an injury. (**B**): Differences in mean NLR (log 10 transformed) as a function of rearing and presence of a dependent infant (i.e., infant younger than 6 months of age). MR: mother-reared, NR: nursery-reared.

**Table 1 vetsci-11-00423-t001:** Studies with neutrophil to lymphocyte ratio as a dependent variable in nonhuman primates.

Citation	Species	Age	Sex	Findings
[17]	*Macaca fascicularis*	3–4 years	5 males	Increased NLR (as well as cortisol) following 15 h of air and truck transport from China to Korea. Returned to baseline 1 week after arrival.
[18]	*Macaca mulatta*	3–4 years	10 females	Increased NLR (as well as cortisol) following chair-restraint training. Returned to baseline after 3 weeks.
[19]	*Macaca mulatta*	mean age = 1.9 years	27 males	Higher NLR following relocation to a new housing area. Higher NLR in monkeys with a short-version serotonin allele (higher emotionality monkeys) compared to a long-version allele (normal emotionality monkeys).
[14]	*Pan troglodytes*	3–31 years	19 males, 20 females = 30	Higher NLR with higher BMI and older age.
[11]	*Pan troglodytes*	2–58 years	185 males, 225 females = 410	Longitudinal data: no change within individuals over a 10-year timespan.
				Cross-sectional data: NLR highest in middle-aged individuals. Higher NLR in males and mother-reared individuals.
				Mortality data: individuals with higher NLRs died at younger ages.
[15]	*Macaca mulatta*	88–134 days	2071 males, 2506 females = 4557	Lower NLRs in indoor-reared, SPF, and male individuals.
				Lower NLR was associated with higher stress values, emotionality, later risk for airway hyperresponsiveness, and diarrhea.
[12]	*Papio anubis*	0–19 years	159 males, 228 females = 387	Higher NLR in females, mother-reared individuals, and young adult individuals.
				NLR was higher during pregnancy and following transport to a new facility.
				Transport stress NLR was heritable, while routine NLR was not heritable.
Current study	*Papio anubis*	0–21 years	284 females, 233 males = 517	Longitudinal data: no significant change within individuals in NLR over a 5-year timespan. However, females, juveniles, and young adults show an increase in NLR over time.
				Cross-sectional data: significant positive correlation between sedation order and NLR. Baboons with higher sedation rates per month exhibited lower NLRs.

**Table 2 vetsci-11-00423-t002:** Raw neutrophil to lymphocyte (NLR) means, standard deviations, and sample sizes (N) across the two datasets used in the current study.

Dataset/Study	Age Category	Sex	Mother-Reared			Nursery-Reared		
			Mean NLR	SD	N	Mean NLR	SD	N
1: Longitudinal NLR and sedation rate (N = 532)	**Juvenile (0–4 years)**	Male	3.42	2.95	122	1.49	1.05	90
		Female	4.94	3.74	72	1.80	1.35	89
		**Total**	3.99	3.33	194	1.65	1.21	179
	**Young Adult (5–9 years)**	Male	2.68	0.67	5	2.48	1.16	7
		Female	4.37	1.61	32	3.59	1.81	20
		**Total**	4.14	1.62	37	3.30	1.71	27
	**Older Adult (10–14 years)**	Male	4.34	1.06	4	3.12	0.65	4
		Female	3.82	1.88	18	3.92	1.49	28
		**Total**	3.91	1.75	22	3.82	1.43	32
	**Geriatric (≥15 years)**	Male	n/a	n/a	n/a	5.22	0.08	2
		Female	5.04	3.31	3	4.24	2.27	36
		**Total**	5.04	3.31	3	4.29	2.22	38
2: NLR at physical exam, sedation order, and health (N = 231)	**Juvenile (0–4 years)**	Male	5.29	4.16	45	3.06	2.03	9
		Female	6.46	4.37	42	4.63	3.70	32
		**Total**	5.85	4.28	87	4.28	3.44	41
	**Young Adult (5–9 years)**	Male	4.72	4.02	3	5.22	3.65	4
		Female	9.88	6.62	32	5.98	5.36	26
		**Total**	9.31	6.55	35	5.82	5.00	30
	**Older Adult & Geriatric (≥10 years)**	Male	3.29	0.58	3	4.39	2.68	2
		Female	7.18	4.50	18	4.75	3.24	15
		**Total**	6.48	4.33	21	4.71	3.11	17

Note: For dataset/study 1, the age category was based on age at the last NLR (i.e., most recent age). For dataset/study 2, the age category was based on age at the timepoint at which NLR was calculated. In dataset 2, older adult and geriatric age categories were grouped together due to low sample size. NLR values presented are raw values. NLR = neutrophil %/lymphocyte %.

**Table 3 vetsci-11-00423-t003:** Analyses within dataset/study 1 and 2, with corresponding statical test and outcome details.

Dataset 1: Longitudinal NLR and Sedation Rate						
Analysis Description	Statistical Test	Sample	Independent Variable(s)	Dependent Variable	Covariate(s)	Result(s)
1a. Longitudinal NLR	Repeated Measures ANCOVA	Non-study baboons with NLR years 1 through 5, N = 174	Sex, rearing	lg10NLR years 1–5	Most recent age	Significant effect of time; pairwise comparisons: no differences between years
1b. Cross-sectional age and NLR	Curve estimation	Non-study baboons, N = 472	Most recent age	Most recent lg10NLR		Significant quadratic and linear relationship
2a. NLR and assignment to study, sedation rate	Linear regression	The entire sample (including study baboons), N = 532	Sex, rearing, and age at the most recent NLR on the first block	Most recent lg10NLR		Age and rearing significant predictors—see Table 3 for coefficients
			Assignment to study on the second block			n.s.
			Sedation rate on the last block			Significant negative relationship
2b. NLR and sedation rate using a matched sample	Linear regression	Baboons matched with study baboons on age, sex, and rearing, N = 131 (66 study and 65 non-study)	Sex and rearing on the first block	Most recent lg10NLR		Replicated result 2a
			Assignment to study on the second block			Replicated result 2a
			Sedation rate on the last block			Replicated result 2a
2c. NLR change over time as a function of sedation rate	Linear regression	Non-study baboons with a minimum of 24 months between timepoints, N = 237	Sex, rearing, and most recent age on the first block	Change score (normally distributed)		Sex and age were significant predictors of NLR change scores.
			Sedation rate on the last block			n.s.
**Dataset 2: NLR and Sedation Order, Health Parameters**						
3. NLR and sedation order	Linear regression	Entire sample, N = 231	Sex, rearing, and age on the first block	lg10NLR		Sex and rearing were significant predictors
			Sedation order on the second block			Sedation order was a significant predictor
4. NLR and injury	Univariate ANCOVA	Entire sample, N = 231	Sex, rearing	lg10NLR	Age	The main effects of sex and rearing were significant
			Injury (y/n)			Injury n.s., but trending injury by sex interaction
5. NLR, pregnancy, dependent infant	Univariate ANCOVA	Adult females only, N = 112	Rearing	lg10NLR	Age	The main effect of rearing was significant
			Pregnancy (y/n)			n.s.
			Dependent infant (present/absent)			Trending infant by rearing interaction

Note: n.s.: not significant.

**Table 4 vetsci-11-00423-t004:** Regression statistics and coefficients for models predicting log-transformed NLR (lg10).

	B Value	Standard Error	Beta	t Value	*p* Value
Intercept	0.534	0.121		4.396	0.000
Age *	0.074	0.029	0.256	2.585	0.011
Sex	−0.109	0.060	−0.128	−1.802	0.074
Rearing *	−0.226	0.099	−0.169	−2.283	0.024
Study assignment	0.084	0.089	0.099	0.950	0.344
Sedation rate *	−0.416	0.092	−0.486	−4.502	0.000

Note: * denotes significant effect in the model.

## Data Availability

Data are available from the corresponding author upon reasonable request.

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
