# Peer review of "Longitudinal Baboon (Papio anubis) Neutrophil to Lymphocyte Ratio (NLR), and Correlations with Monthly Sedation Rate and Within-Group Sedation Order"

_vetsci, 2024, doi:10.3390/vetsci11090423_

Round 1
Reviewer 1 Report
Comments and Suggestions for Authors
Dear authors,
Stress control is an important issue for captive non-human primate (NHP) welfare, also stress can affect behavioral and social studies in NHP. Many factors can induce NHP stress, such as transportation, restrains, moving to new place. Neutrophil to Lymphocyte Ratio (NLR) has been considered as a marker of stress. Sedation, a frequently used method for NHP manipulation, is also thought as inducer of stress. So far, there is no report on how sedation rate and order within a group affect baboon NLR.
Your work provided us valuable information concerning baboon NLR in an over 500 colony, covering 0-20.5 years old animals. In particular, offering information about sedation effect on NLR, which is especially valuable for behavioral research requiring frequent sedation.
In addition, the you also offer us for the first time, so far as I know, how NLR changes over age and between gender, as well as acute and chronic stress in baboon.
I believe your results are good reference for researcher and animal welfare officer in this field.
Thank you for sharing!
Best!
Reviewer 2 Report
Comments and Suggestions for Authors
Dear authors:
NLR is a commonly used parameter to evaluate stress and the prognosis of some diseases in various species. This document provides novel information because these evaluations were carried out at NHPs, therefore it is an interesting study with great value for science. I would like to congratulate the authors for the preparation of this document, since I consider it is well prepared, with an important sample of individuals and because it provides very valuable information. However, I have added some comments and suggestions in order to improve the document.
Line 61: Please add a brief description of what it is NLR and how is evaluated and interpretated. For example, mention the normal values ​​expected in the NLR in other species and in NHPs. I can see that you describe it later in lines 105-129, but you do not mention the expected parameters.
Line 97: Please erase “; Neal et al., 2024)”.
Line 168: In material and method section, please briefly describe how you do blood sample and describe the materials an equipment that were used to do it.
Line 306, Figure 1: I consider that adding the significant difference (if there was one) in the graphs through literals could improve figure 1. The same goes for figure 4 (line 360).
Line 441-448: Perhaps this information would be better in the material and methods section.
References: Please complete references 1 and 16.
Reviewer 3 Report
Comments and Suggestions for Authors
GENERAL COMMENTS:
The authors examined baboon NLR longitudinally, considering its relationship with health parameters and whether NLR was affected by sedation rate, as well as the order of sedation within a group. The study is very interesting and suitable for the Veterinary Sciences audience, with important results for the community. However, before considering the manuscript for acceptance, I have some concerns related to the Introduction section and the objectives.
The Introduction is currently presented in a manner that lacks fluidity and coherence, which detracts from the overall quality of the manuscript, especially when compared to the more cohesive Results and Discussion sections. While the Introduction contains important information, it is delivered in a lengthy and overly exhaustive manner, more reminiscent of a systematic review, especially with the inclusion of details such as the number of studies found and their publication dates. This approach muddles the focus of the Introduction, making it difficult to discern a clear narrative thread. Additionally, Table 1 should be removed from this section, as it does not contribute meaningfully to the flow of the introductory content.
The objectives of the study are also presented in a disjointed manner, appearing twice in the text (Lines 162-167 and Lines 145-146) without a clear connection. I strongly recommend a complete rewrite and restructuring of the Introduction to clearly present the guiding questions of the study, such as: What is the Neutrophil to Lymphocyte Ratio (NLR)? Why is studying it in nonhuman primates important? What is already known about NLR in NHPs from the literature? This section should culminate in a final paragraph stating the study's hypotheses and objectives, both general and specific. The authors are encouraged to review and revise the manuscript accordingly. Detailed comments and suggestions for further improvement are provided below.
SPECIFIC COMMENTS:
Please consider italicizing the Latin expressions 'e.g.' and ‘i.e.’ (Lines 10, 17). Review the manuscript and adjust accordingly. Ensure that you have added the meanings of acronyms (e.g., NLR, MR, NR, etc.) to the captions of ALL FIGURES AND TABLES.
KEYWORDS:
The keywords should differ from those presented in the title. Please adjust accordingly.
INTRODUCTION:
The sentence in Lines 44-49 is quite long and could be shortened for better readability.
The information in Lines 59-60 might be more effective if integrated into the study’s objectives. Please adjust accordingly.
The paragraph in Lines 62-71 reads more like a systematic review. I suggest reworking it to highlight the main findings from the literature in a clearer and more engaging way.
The explanation about low and high NLR values indicating poor health (Lines 71-74) is somewhat unclear. Please clarify how these values correlate with health.
The reference to “explanations” in Lines 99-100 is vague—please specify what’s being explained.
The phrase “increased cortisol reduces lymphocyte count” in Lines 108-110 seems confusing. If cortisol lowers lymphocyte counts, wouldn’t that increase the NLR? Please clarify this point.
Table 1: Consider removing Table 1 from the Introduction section, as it doesn’t seem to add much value here.
The section in Lines 133-146 feels more suited to the Discussion than the Introduction—consider moving it.
The study’s objectives need to be clearer. They’re mentioned twice in the text (Lines 162-167 and 145-146) but not in a cohesive way. Please clarify.
MATERIALS AND METHODS:
When presenting the sex ratio, please include the numbers for both males and females (Line 170).
Table 2: It would help to add a line separating MR and NR; include explanations for the acronyms NLR, MR, and NR; in the footnote, clarify if NLR values were transformed or not [e.g., N(%) / L(%)]; specify the correct unit of measurement for NLR in the column title; remove the bold formatting from the gender titles (Male and Female).
Why are 'Older Adult & Geriatric' animals grouped together in Study 2, but separated in Study 1? Are these different datasets?
Table 3: Provide explanations for ‘IV(s)’ and ‘DV’.
It would be better to present statistical test results in the Results section rather than in Table 3.
Close the parenthesis in Line 272.
Please include a phrase stating the p-values used to determine significance and trends in the methodology.
RESULTS:
Explain the values in parentheses following ‘F’ (Lines 300, 302, 303).
Figure 1: It’s unclear what Figure A represents—does it show the NLR(log) adjusted/simple means? Add the meaning of the NLR acronym.
Figure 2: Clarify what the three lines represent and add legends; add an explanation for the NLR acronym.
Figure 3: Add the meaning of the NLR acronym. Clarify what the continuous and dotted lines mean.
Since sex influences NLR, I suggest the authors consider adding a breakdown by sex in Figure 2 (or also maybe Figure 3). It could keep the same layout but use different colors to represent each sex. Please be sure to use a color-blind-friendly palette.
Figure 4: Add an explanation for the NLR acronym.
DISCUSSION:
It might be helpful to add a paragraph at the end of the Discussion about the study’s limitations. You could include the points from Lines 133-146 here. Also, consider moving the suggestions for future studies from Lines 569-583 into the Discussion.
CONCLUSIONS:
The Conclusions should clearly address each of the study’s objectives. Please revise the objectives and make any necessary adjustments to the Conclusion section.
Comments on the Quality of English Language
No additional comments.
Reviewer 4 Report
Comments and Suggestions for Authors
Excellent manuscript. Very interesting potential additional parameter to assess in relation in NHP welfare.
Author Response
Thank you for your review of our manuscript.
Round 2
Reviewer 3 Report
Comments and Suggestions for Authors
After review and the incorporation of the information, I find that my concerns regarding the manuscript have been addressed.